# Improvement of Glycemic Control by a Functional Food Mixture Containing Maltodextrin, White Kidney Bean Extract, Mulberry Leaf Extract, and Niacin-Bound Chromium Complex in Obese Diabetic *db/db* Mice

**DOI:** 10.3390/metabo12080693

**Published:** 2022-07-26

**Authors:** Huei-Ping Tzeng, Chen-Yuan Chiu, Shing-Hwa Liu, Meng-Tsan Chiang

**Affiliations:** 1Institute of Toxicology, College of Medicine, National Taiwan University, Taipei 10051, Taiwan; hptzeng811@gmail.com; 2Center of Consultation, Center for Drug Evaluation, Taipei 11575, Taiwan; kidchiou@gmail.com; 3Department of Pediatrics, College of Medicine and Hospital, National Taiwan University, Taipei 10051, Taiwan; 4Department of Medical Research, China Medical University Hospital, China Medical University, Taichung 40402, Taiwan; 5Department of Food Science, National Taiwan Ocean University, Keelung 20224, Taiwan

**Keywords:** functional foods, steady-fiber granule, type 2 diabetes mellitus, insulin resistance, hyperglycemia

## Abstract

Steady-fiber granule (SFG) is a mixture containing maltodextrin, white kidney bean extract, mulberry leaf extract, and niacin-bound chromium complex. These active ingredients have been shown to be associated with improving either hyperglycemia or hyperlipidemia. This study was undertaken to evaluate the potential of SFG in the regulation of blood glucose homeostasis under obese diabetic conditions. Accordingly, *db/db* mice (8 weeks old) were administered with SFG at doses of 1.025, 2.05, or 5.125 g/kg BW daily via oral gavage for 4 weeks. No body weight loss was observed after SFG supplementation at all three doses during the experimental period. Supplementation of SFG at 2.05 g/kg BW decreased fasting blood glucose, blood fructosamine, and HbA1c levels in *db/db* mice. Insulin sensitivity was also improved, as indicated by HOMA-IR assessment and oral glucose tolerance test, although the fasting insulin levels were no different in *db/db* mice with or without SFG supplementation. Meanwhile, the plasma levels of triglyceride were reduced by SFG at all three doses. These findings suggest that SFG improves glycemic control and insulin sensitivity in *db/db* mice and can be available as an option for functional foods to aid in management of type 2 diabetes mellitus in daily life.

## 1. Introduction

The prevalence of obesity and type 2 diabetes mellitus (T2DM) has increased globally during the last century, both of which have been recognized as worldwide epidemics, raising the risk of cardiovascular disease by nearly twofold [1]. The International Diabetes Federation (IDF) estimated that 537 million adults are living with diabetes (one out of 10) worldwide in 2021, and this number is projected to rise to 643 million by 2030 [2]. T2DM accounts for 90% to 95% of all diabetes cases, and approximately 90% of the T2DM patients in USA are overweight or obese, defined as a body mass index (BMI) of 25 kg/m^2^ or higher [3]. Obesity is regarded as the major risk factor for T2DM development [4]. In addition to genetic factors, environmental factors, such as increased fat/caloric intake and decreased physical activity, contribute to the development of obesity and insulin resistance, followed by progression to impaired glucose tolerance and ultimately development of T2DM [5]. It is commonly believed today that lifestyle changes, such as adopting a healthier diet (calorie restriction/higher fiber intake) combined with moderate-intensity exercise, may prevent, delay, or manage T2DM and possibly the related complications [6,7].

Numerous studies show that higher intakes of dietary fiber or whole grains are associated with a reduction in the risk of mortality and incidence of a wide range of noncommunicable diseases, including T2DM [8]. The National Academies gave nutrition recommendations of an intake of 38 and 25 g per day of total fiber for adult men and women respectively [9]. Currently, most people worldwide consume around or less than 20 g of dietary fiber per day, which are lower than recommended [10,11]. In terms of resistant starch (RS), as a source of dietary fiber, the estimated daily intake by Americans is approximately 3 to 8 g per person, which is lower than the recommended daily intake for providing health benefits [12]. RS is defined as any starch that escapes digestion by amylases in the small intestine and passes to the colon to be fermented by microbiota [13]. Intake of 6 to 12 g RS per meal is manifested to have beneficial effects on postprandial glucose and insulin levels, and intakes of ~20 g per day are considered necessary to have positive impacts on gastrointestinal health, such as promoting healthy fecal bulk [12]. This, thus, emphasizes the need to increase the dietary fiber or RS content of daily diets. Given that most of the Western diet contains little dietary fiber or RS, incorporation of functional foods made with added dietary fiber or RS into the daily diet could be an alternative approach to deliver health benefits for diabetic patients in regulation of blood glucose and body weight, which may help manage the progression of T2DM [7,14].

Steady-fiber granule (SFG), a functional food mixture registered in Taiwan, is formulated with four major active ingredients: 88% resistant maltodextrin (RMD), 5% white kidney bean (*Phaseolus vulgaris*) extract (WKBE), 0.2% mulberry leaf (*Morus alba* L.) extract (MLE), and 0.02% niacin-bound chromium (NBC) complex. The health benefits of each ingredient have been reported. RMD is known to be a water-soluble, non-viscous dietary fiber made from corn starch and classified as RS type V, with the capacity of maintaining health intestinal regularity, improving glucose and lipid metabolism, and promoting satiety [15,16]. WKBE, containing α-amylase inhibitors, is considered to limit carbohydrate digestion and absorption, along with beneficial effects on body weight, glycemic control, and metabolic health [17,18]. MLE has a long history of use as a traditional medicine, demonstrated to have wide range of biological activities [19], which includes decreasing digestion and absorption of starch [20], and reducing postprandial hyperglycemia, mainly via inhibiting α-glucosidase [21,22]. Trivalent chromium is an essential mineral, and NBC is a safe and bioavailable form of chromium [23], which is involved in enhancing insulin sensitivity and maintaining normal blood glucose levels, with the ability to lower plasma cholesterol levels and blood pressure, as well as inhibit oxidative stress [24,25].

T2DM is characterized by hyperglycemia resulting from insulin deficiency caused by pancreatic β-cell dysfunction and insulin resistance in insulin-sensitive tissues such as liver, muscle, and adipose tissue [6]. While stimulated by elevated blood glucose levels following a meal, insulin secreted by β-cells will work to maintain normal blood glucose mainly by promoting the uptake of glucose in skeletal muscle and adipose tissue, as well as by suppressing glucose efflux from liver and lipolysis from adipose. Interaction among all of those organs and tissues involved ensures glucose homeostasis [26]. Impairments in insulin secretion and insulin sensitivity may result in impaired glucose tolerance, leading to T2DM [5]. Thus, improvement of insulin secretion or sensitivity may, therefore, improve glucose homeostasis, which is beneficial in the development and management of T2DM. SFG has wide spectrum of safety manifested in our previous study [27], and its major ingredients, as mentioned above, are all known to have beneficial effects on obesity or diabetes; thus, we hypothesized that a dietary intervention of the functional food SFG would improve the glycemic control and blood lipids under obese diabetic conditions. Accordingly, *db/db* obese diabetic mice at age of 8 weeks were orally administered with SFG for 4 weeks to observe whether SFG had beneficial effects on obesity and diabetes by attenuating overweight, hyperglycemia, hyperinsulinemia, and dyslipidemia in the obese diabetic model.

## 2. Materials and Methods

### 2.1. Animals and Diets

Steady-fiber granule (SFG) was provided by the Sinphar Pharmaceutical (Yilan, Taiwan). As described previously [27], SFG powder contains digestion-resistant maltodextrin (88%), white kidney bean (*P. vulgaris*) extract (5%), mulberry leaf (*Morus alba* L.) extract (0.2%), and niacin-bound chromium complex (0.02%), accounting for 93.22% of the total weight, while the remaining 6.78% constitutes inert excipients to add flavors to the product. SFG was dissolved in distilled water to make 0.5125 g/mL stock solution, which was then diluted to become 0.205 g/mL and 0.1025 g/mL before performing the tests.

All animal experiments were approved by the Animal Research Committee of the National Taiwan University College of Medicine (IACUC Approval No. 20150334) and carried out in strict accordance with the regulations of Taiwan and NIH guidelines on the care and welfare of animals. All the studies were performed in male C57BL/6J, as lean control mice and leptin-receptor mutant (*db/db*) mice. The *db/db* strain mice were used as a rodent model of obese diabetes, mimicking the condition of human T2DM. This mouse, with a defect in the gene coding the leptin receptor, is characterized by leptin resistance, which leads to hyperphagia, obesity, insulin resistance, and consequently, T2DM [28,29]. Therefore, it makes *db/db* mice a good rodent model to evaluate whether a test compound has antihyperglycemic or insulin-sensitizing potential.

The mice were obtained from the National Laboratory Animal Center (Taipei, Taiwan) and maintained under controlled temperature (22–24 °C), with a relative humidity of 40–60%, on a 12 h light/dark lighting cycle. Following the schedule shown in Figure 1, mice at age of 6 weeks were ad libitum given rodent chow (LabDiet 5001, PMI Nutrition International, St. Louis, MO, USA) and water for 2 weeks, and then randomly divided into five groups with eight mice in each group. Age-matched C57BL/6J mice were administered vehicle (distilled water) via oral gavage, and *db/db* mice were administered distilled water or SFG at doses of 1.025, 2.05, and 5.125 g/kg BW via oral gavage once daily for 4 weeks; these mice were expressed as *db*/*db*, SFG 1.025, SFG 2.05, and SFG 5.125, respectively. The doses of SFG were chosen on the basis of our previous studies and showed no signs of any toxicity tested [27]. In brief, the recommended dosage of SFG for human is one serving (5 g) to two servings per day. According to the body surface area, converting human dose to an equivalent mouse dose involves multiplying the human dose by 12.3 [30]. Therefore, the doses for mice were set at 1.025 g/kg as onefold dose, 2.05 g/kg as twofold dose (high recommended dose), and 5.125 g/kg as fivefold dose. Body weight, food intake, and water consumption were monitored weekly. Body weight was expressed as the average body weight per mouse; food and water intake were calculated as the average amount per day per mouse. On the last day of the experiment after 12 h fasting, mice were anesthetized, and blood samples were obtained from the retro-orbital sinus using heparin-coated capillary tubes for blood glucose measurement. At the end of experiments, blood samples were subsequently taken by cardiac puncture, and plasma was collected by centrifugation (1570× *g*, 20 min, 4 °C) and stored at −20 °C for future determination of various biochemical parameters.

### 2.2. Measurement of Fasting Plasma Glucose and Oral Glucose Tolerance Test (OGTT)

Blood sampling from retro-orbital sinus was used for measuring the fasting glucose levels, which were detected by a Glucose Enzymatic Kit (Audit Diagnostics, Cork, Ireland). Then, the OGTT was performed. Glucose load (1.5 g/kg BW) was administered by oral gavage to mice, and plasma glucose levels were measured at 0 (before glucose challenge), 30, 60, 90, and 120 min post administration. The area under the curve (AUC) of the OGTT was calculated using Prism 6 (GraphPad Software, La Jolla, CA, USA) as a measure of glucose tolerance.

### 2.3. Measurement of Insulin and Homeostatic Model Assessment for Insulin Resistance (HOMA-IR)

The plasma samples collected from mice were used for measuring insulin levels, which were detected using the Mouse Insulin ELISA kit (Mercodia AB Inc., Uppsala, Sweden). The fasting glucose and insulin data were then used to generate the HOMA-IR with the following formula [31]: HOMA-IR = (fasting blood glucose (mg/dL) × fasting insulin (mU/L))/405.

### 2.4. Measurement of Blood Biochemical Parameters

Plasma levels of aspartate aminotransferase (AST) and alanine aminotransferase (ALT) were measured spectrophotometrically using commercial kits (RANDOX Ltd., Antrim, UK) to determine hepatotoxicity. Concentrations of triglyceride (TG) and total cholesterol (TC) were determined using colorimetric enzymatic kits (Audit Diagnostics, Cork, Ireland). Glycated hemoglobin (HbA1c) was detected by hematology analyzer (SYSMEX, Copenhagen S, Denmark). Fructosamine was estimated using a Fructosamine Kit (Hospitex diagnostics Lp, League City, TX, USA). Fructosamine (μmol/L) = (Sample_5 min_ − Sample_0 min_)/(Calibrator_5 min_ − Calibrator_0 min_) × 365.

### 2.5. Statistical Analysis

Statistical analysis was performed using Prism 6 (GraphPad Software, La Jolla, CA, USA). One-way analysis of variance (ANOVA) was used for multiple comparisons, and post hoc analysis was performed with Duncan’s multiple range test or Tukey test. Results were shown as the mean ± SD and considered significant if *p*-values were <0.05.

## 3. Results

### 3.1. Effects of SFG on Weight Gain, Food Intake, and Water Consumption in db/db Mice

As expected, the food and water intake, as well as the body weight (BW), was significantly increased in *db/db* mice compared to control lean mice during the experimental period (Table 1). In *db/db* mice, the BW was increased significantly from week 1 to week 4 (35.2 g to 40 g), whereas control mice maintained a steady BW increase (23.0 g in week 1 to 24.3 g in week 4). Supplementation with SFG at three doses had no effect on BW gain. Furthermore, the food and water intake was similar in *db/db* mice with or without SFG supplementation (Table 1). Overall, SFG supplementation did not alter BW, food intake, and water consumption in *db/db* mice from 8 weeks of age to 12 weeks of age.

### 3.2. SFG Improved Glycemic Parameters in db/db Mice

Increased fasting blood glucose, glycated hemoglobin (HbA1c), and blood fructosamine levels were observed in *db/db* mice. The fasting blood glucose at the end of experiment was 358 ± 59.5 mg/dL in *db/db* mice compared to 125 ± 21.1 mg/dL in control mice. Supplementation of SFG at 2.05 and 5.125 g/kg reduced the fasting blood glucose levels significantly by 25% and 18.6%, respectively, compared to those in *db/db* mice (Figure 2A). SFG at 2.05 g/kg also distinctly decreased the levels of HbA1 by 18.7% (Figure 3A), which is a long-term indicator of glycemic control, revealing the mean plasma glucose over the past 2 or 3 months [32]. Moreover, fructosamine, formed by a nonenzymatic glycation of blood protein, reflects shorter-term glucose control within the previous 2–3 weeks [33]. SFG supplementation at three doses reduced plasma fructosamine levels by 34%, 49%, and 40%, respectively (Figure 3B), showing strong evidence of an SFG effect against hyperglycemia provoked in *db/db* mice. In addition, we measured the plasma insulin levels. Although the fasting glucose levels were better in *db/db* mice supplemented with SFG at 2.05 and 5.125 g/kg, SFG at all three doses had no effect on fasting insulin levels, as compared to these of *db/db* mice (Figure 2B).

### 3.3. SFG Improved Glucose Uptake by Enhancing Insulin Sensitivity in db/db Mice

Given that hyperglycemia was partially rescued by SFG supplementation, we assessed whether this benefit of SFG is dependent on gaining insulin sensitivity. From this perspective, we first performed the HOMA-IR analysis to evaluate the status of insulin action. As shown in Figure 2C, the HOMA-IR value was 35% lower in *db/db* mice fed with 2.05 g/kg SFG compared to *db/db* mice, which suggests that supplementation of SFG may enhance glucose disposal in *db/db* mice.

OGTT was then implemented to further assess the insulin secretion and action [34]. In control mice, the concentration of blood glucose reached a peak at 30 min and returned to a normal level within 2 h, showing glucose homeostasis (Figure 4A). In contrast, the *db/db* mice showed the highest glucose levels throughout the measuring points, indicating a typical phenotype of insulin resistance. SFG-supplemented mice at doses of 1.025 and 2.05 g/kg showed a dramatic decrease in postprandial blood glucose levels by 22.7% and 32.1%, respectively, at 120 min post glucose challenge, as compared to those in *db/db* mice. Furthermore, the AUC calculated from OGTT was reduced by 28% after SFG supplementation, from 56,949 mg/dL·min in *db/db* mice to 40,723 mg/dL·min in SFG 2.05 *db/db* mice (Figure 4B). The diminished AUC further proved that SFG supplementation enhanced insulin sensitivity, allowing glucose to translocate from the blood into the cells more efficiently, thus improving the impaired glucose tolerance in *db/db* mice. As such, our findings suggested that SFG has the ability to gain insulin sensitivity and improve glucose uptake under obese diabetic conditions.

### 3.4. SFG Improved Plasma Triglyceride (TG) Levels in db/db Mice

We then assessed whether SFG had effects on the insulin resistance-associated lipid profile, including total cholesterol (TC) and TG. The *db/db* mice showed the development of hypertriglyceridemia with plasma TG levels up to 192 mg/dL, while SFG supplementation at 1.025, 2.05, and 5.125 g/kg BW prominently diminished the levels of TG by 17.6%, 12.2%, and 16.8%, respectively, as compared to the *db/db* mice (Table 2). However, the suppressive effect of SFG did not exhibit a dose-dependent profile. SFG supplementation at 1.025 g/kg BW, but not at 2.05 and 5.125 g/kg BW, led to a significant decrease in increased cholesterol levels in *db/db* mice (Table 2).

Plasma concentrations of aspartate aminotransferase (AST) and alanine aminotransferase (ALT) were examined as indicators of hepatic injury. The levels of plasma AST in all groups showed no significant difference at the end of experiment, although the plasma AST levels were increased, but not statistically significant, in *db/db* mice (Table 2). Moreover, the plasma ALT levels were significantly increased in *db/db* mice compared to control mice; however, the ALT levels among the *db/db* and SFG-treated groups were not significantly different (Table 2).

## 4. Discussion

T2DM is a glycemic disorder with the diagnosis mainly based on measurements of fasting plasma glucose, 2 h OGTT, and HbA1c. Among them, the HbA1c correlates with overall glycemia as it reflects the average glucose over 2–3 months [35]. Our results showed that a treatment of SFG at 2.05 g/kg BW for 4 weeks significantly decreased the fasting and post-load blood glucose and HbA1c levels in *db/db* mice. Meanwhile, SFG at all three doses tested (1.025–5.125 g/kg BW) had inhibitory effects on fructosamine levels, which reflects the average glucose concentrations over 2–3 weeks. The data of fructosamine levels provided more vital information than HbA1c in our study, as *db/db* mice received SFG for 4 weeks only. The results, thus, demonstrate that SFG exerts a sustained antihyperglycemic effect. The ability of SFG to reduce hyperglycemia can be attributed to its insulin-sensitizing activity, since SFG 2.05 *db/db* mice displayed a better responsiveness to insulin and a better glucose disposal as evidence of a lower HOMA-IR score and a smaller AUC of 2 h OGTT compared to *db/db* mice. Moreover, dyslipidemia was also observed in *db/db* mice, and SFG reduced plasma TG levels, whereas TC levels remain unaffected. Furthermore, no effects of SFG on BW control was observed during the study period. Dietary fiber intake has been reported to decrease food intake and prevent obesity [14]; however, SFG had minimal but not significant effects on food intake, suggesting the study period might have been too short to observe a weight change. A longer-term study will be necessary to address this weight loss effect of SFG in the future.

The mechanism via which SFG acts to ameliorate insulin resistance and improve glycemic control is undetermined and needs to be further studied; however, the individual components of SFG have been investigated. As mentioned previously, the ingredient with the greatest amount of weight in SFG is RMD. Being a soluble RS, RMD is believed to be highly resistant to digestive enzymes in the small intestine and then fermented to short-chain fatty acids (SCFAs), principally acetate, propionate, and butyrate, in the colon whereon these SCFAs are absorbed by colonocytes and mediate several local and systemic functions, including triggering the secretion of satiety hormones such as glucagon-like peptide 1 (GLP-1) and peptide YY (PYY), thus modulating the immune response and acting as mediators for the complex microbiota-gut-brain communication [36]. These properties contribute to its antidiabetic and anti-obesity effects by lowering plasma glucose and lipid, attenuating pancreas damage, reducing energy intake, and promoting satiety [15,37]. In other words, RS or RMD acts as a prebiotic encouraging the growth and diversity of healthy host bacteria, allowing the production of SCFAs and stimulating the release of gut hormones [38]. Thus, the addition of RMD to the diet as a bioactive component of the functional food SFG is an effective way to increase gut microbiota and aid in glycemic control and possibly weight control. Fibersol-2^TM^, which is also an RMD from corn fiber, has been reported to decrease hunger and energy intake, and increase satiety hormones in humans when ingested with a meal [39]. Additionally, RMD from tapioca starch has been shown to have benefits on glycemic control by reducing incremental plasma glucose and serum insulin in health human subjects [40]. However, high intake of RS or RMD may cause gastrointestinal (GI) discomfort due to excess gas production [15].

So as to minimize the adverse effect on the GI system, the use of RMD should be limited. Therefore, a combination of other carbohydrate blockers with RMD into a diet is a viable approach to aid in regulating glucose homeostasis. Carbohydrates, before being absorbed into the body, must be degraded into monosaccharides, which involves two major enzymes, namely, amylase and glucosidase [41]. Inhibition of enzymes responsible for carbohydrate digestion is a way to slow their absorption. It has been reported that Phase 2^®^ Carb Controller, an α-amylase inhibitor from common white bean, reduces the rate of absorption of carbohydrates, thereby decreasing calorie intake and promoting weight loss, as well as reducing the postprandial spike in blood glucose levels [17]. Furthermore, several products from MLE appear to effectively lower the postprandial plasma glucose and improve insulin resistance [42]. Studies have shown that the most bioactive constituents of MLE responsible for its antidiabetic activities are believed to be 1-deoxynojirimycin (DNJ), which is a d-glucose analogue and works as a potent α-glucosidase inhibitor by competitively binding to the active site of the enzyme [43]. Therefore, the combination of WKBE and MLE, by inhibiting both α-amylase and α-glucosidase, may additively retard carbohydrates uptake and suppress postprandial plasma glucose. Altogether, in addition to containing RMD, it is reasonable to include WKBE and MLE in the composition of SFG to potentiate the favorable effects of RMD on glycemic control, via further blocking or slowing digestion and absorption of dietary carbohydrate in the GI tract.

MLE displays various beneficial effects on regulation of diabetes, and the molecular basis underlying these effects has long been studied. In addition to limiting carbohydrate digestion and absorption, MLE exerts antidiabetic activity in *db/db* mice through stimulating glucose disposal in skeletal muscle cells via the PI3K/Akt and AMPK pathways [44]. MLE increases glucose uptake via activating the PI3K/Akt pathway and the translocation of glucose transporter 4 (GLUT4) to the plasma membrane in rat adipocytes [45]. MLE restores arterial pressure in streptozotocin-induced chronic diabetic rats [46]. MLE also exerted anti-inflammatory effects against LPS-induced inflammation through an AMPK/Nrf2 signaling pathway in A549 alveolar basal epithelial cells [47].

The active ingredient present in the smallest amount in SFG is NBC, whose average intake for adults is generally less than the recommended daily intake [48]. A reference daily intake for Cr was set at 120 μg/day by the Food and Drug Administration [49]. Supplementation of bioavailable Cr as functional foods is adequate to meet the minimum suggestion of 50 μg/day; it was declared by the National Research Council that the estimated safe and adequate daily dietary intake for Cr ranges from 50 to 200 μg/day [50]. Of note, Cr deficiency leads to diabetic-like signs and symptoms, such as impaired glucose intolerance, hyperinsulinemia, elevated percent body fat, and peripheral neuropathy; Cr supplementation improves these adverse events [48]. The mechanism via which Cr mediates glucose uptake has been proposed [51]. Cr enhances the phosphorylation of insulin receptor β, activates PI3K/AKT insulin signaling, and facilitates the translocation of GLUT4 to the cell membrane. Cr mediates cholesterol efflux from the membranes, which also stimulates GLUT4 translocation and glucose uptake. Moreover, Cr alleviates endoplasmic reticulum stress, rescuing insulin receptor substrate (IRS) from ubiquitination. By directly regulating neurotransmitters, Cr may have potential effects on central control of satiety and energy homeostasis [52]. Hence, NBC, as one of the components, may potentiate the benefits of SFG as a dietary supplement in regulating the blood glucose homeostasis.

Collectively, multiple mechanisms have been proposed to be associated with the benefits of the four bioactive ingredients of SFG in regulation of diabetes and obesity. It is of note that the effects of SFG on hyperglycemia and hypertriglyceridemia are not dose-dependent. SFG at 2.05 g/kg BW had favorable effects on most parameters we tested, whereas SFG at 5.125 g/kg BW only ameliorated fasting glucose levels, fructosamine levels, and TG contents, whose effects were not better than those of the SFG 2.05 group. The benefits of the SFG 2.05 group on improving insulin sensitivity in *db/db* mice were not observed in the SFG 5.125 group; further studies are needed to elucidate this phenomenon. However, in our experience, few functional foods work in a dose-dependent manner. Willis et al. also reported that increasing the consumption of mixed viscous fiber in human subjects did not influence gut hormone levels, ghrelin, and GLP-1 in a dose-dependent manner [53]. They interpreted that different types of fibers may have different impacts on health.

The study had potential limitations. In order to assure accurate and precise dosing, SFG was given to the mice via oral gavage, whose disadvantages include the possibilities of inadvertent tracheal administration and esophageal discomfort. Local inflammation and disturbed absorption may be induced by repeated dosing, which may have affected our results. Furthermore, we only evaluated the effect of SFG for 4 weeks, which might not have been long enough to observe the weight loss effect. Moreover, if we conducted glucose uptake experiments in liver, muscle, or adipose tissue or checked β-cell function, the insulin-sensitizing effect of SFG could have been better described, potentially offering information to explain the different effects on glycemic control between SFG 2.05 and SFG 5.125 group.

## 5. Conclusions

This study clearly illustrated the health benefits of feeding SFG to *db/db* obese diabetic mice. SFG supplementation, together with its wide safety profile, improved hyperglycemia, insulin resistance, and blood TG content with a limited change in body weight in the obese diabetic mice. Given that T2DM is a chronic progressive disease, lifestyle modifications such as healthy dietary interventions, regular physical activity, and weight loss are simple but useful first-line strategies for the management of T2DM [6]. Dietary recommendations worldwide encourage consumption of fiber-rich foods [54]; therefore, supplementation of functional food SFG, to meet the recommended daily intake of RS, is an ideal option for T2DM patients to manage the progression of the disease in their daily life, as it has favorable effects on the regulation of blood glucose homeostasis and lipid profiles under obese diabetic conditions.

## Figures and Tables

**Figure 1 metabolites-12-00693-f001:**
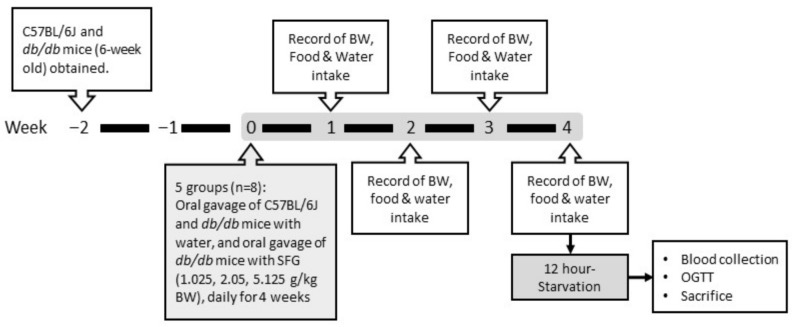
A flowchart of the study showing experimental timeline.

**Figure 2 metabolites-12-00693-f002:**
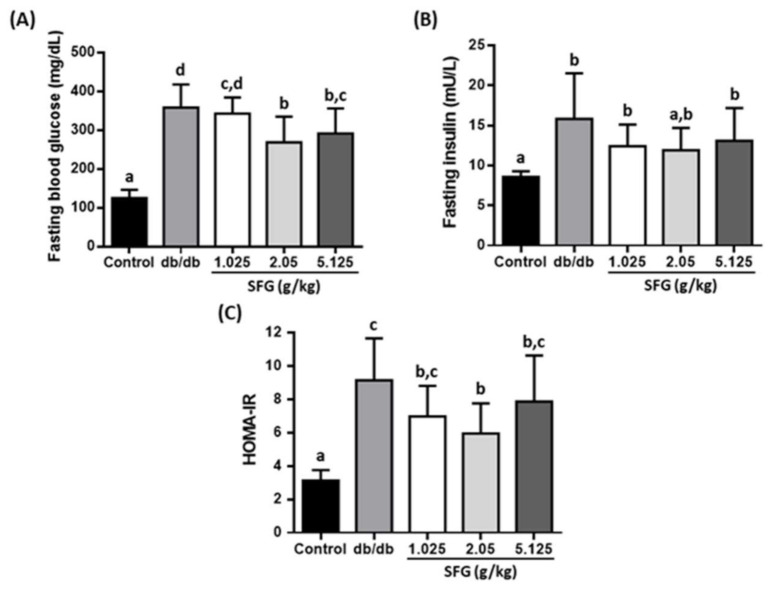
Effects of SFG supplementation on fasting blood glucose, fasting insulin, and insulin sensitivity in *db*/*db* mice. C57BL/6J mice received the vehicle (distilled water) as a control, while *db*/*db* mice received the vehicle and SFG (1.025, 2.05, and 5.125 g/kg BW) for 4 weeks; groups are denoted as *db*/*db*, SFG 1.025, SFG 2.05, and SFG 5.125, respectively. Fasting blood glucose (**A**), fasting plasma insulin levels (**B**), and HOMA-IR (**C**) were then measured after 12 h fasting. Results are expressed as the mean ± SD, *n* = 8. A significant difference is indicated by superscript letters according to one-way with ANOVA post hoc Duncan’s multiple range test. Different superscript letters indicate statistical significance (*p* < 0.05).

**Figure 3 metabolites-12-00693-f003:**
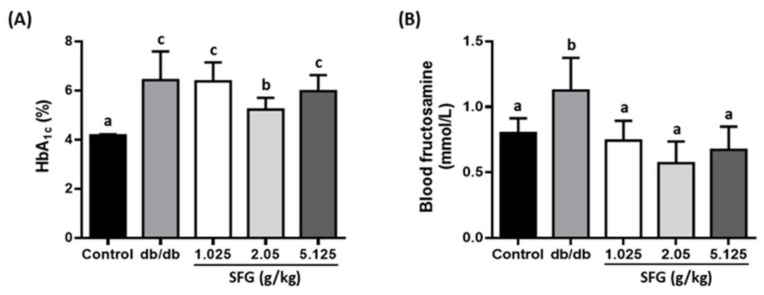
Effects of SFG supplementation on blood fructosamine and HbA1c levels. C57BL/6J mice received the vehicle (distilled water) as a control, while *db*/*db* mice received the vehicle and SFG (1.025, 2.05, and 5.125 g/kg BW) for 4 weeks; groups are denoted as *db*/*db*, SFG 1.025, SFG 2.05, and SFG 5.125, respectively. After fasting for 12 h, a blood sample was collected for the detection of HbA1c (**A**) and fructosamine (**B**) levels. Results are expressed as the mean ± SD, *n* = 8. A significant difference is indicated by superscript letters according to one-way ANOVA with Tukey post hoc test. Different superscript letters indicate statistical significance (*p* < 0.05).

**Figure 4 metabolites-12-00693-f004:**
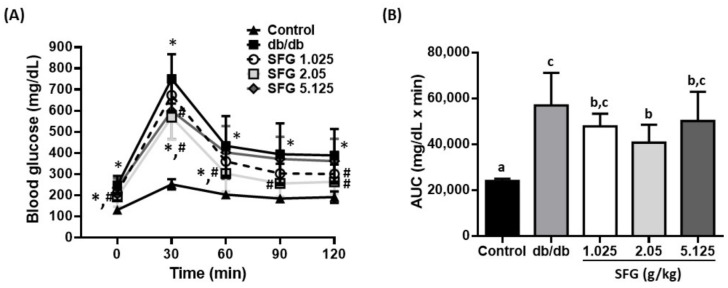
Effects of SFG supplementation on blood glucose levels and area under curve (AUC) during OGTT. C57BL/6J mice received the vehicle (distilled water) as a control, while *db*/*db* mice received the vehicle and SFG (1.025, 2.05, and 5.125 g/kg BW) for 4 weeks; groups are denoted as *db*/*db*, SFG 1.025, SFG 2.05, and SFG 5.125, respectively. After fasting for 12 h, mice were subjected to the oral glucose tolerance test (OGTT). Blood was sampled and examined at 0 (baseline), 30, 60, 90, and 120 min for glucose levels. Results are expressed as the mean ± SD, *n* = 8. * *p* < 0.05 as compared to the control mice; ^#^ *p* < 0.05 as compared to the *db*/*db* mice (**A**). AUC was calculated, and the statistical significance was indicated by superscript letters according to one-way ANOVA with post hoc Duncan’s multiple range test (**B**). Different superscript letters indicate statistical significance (*p* < 0.05).

**Table 1 metabolites-12-00693-t001:** The body weight, food intake, and water consumption of *db*/*db* mice supplemented with different doses of SFG for 4 weeks.

			SFG (g/kg)		
	Control	*db/db*	1.025	2.05	5.125
Body Weight (g)					
Week 1	22.98 ± 0.73 ^a^	35.25 ± 2.87 ^b^	36.48 ± 1.78 ^b^	36.53 ± 2.05 ^b^	36.80 ± 1.65 ^b^
Week 2	24.18 ± 1.09 ^a^	37.13 ± 2.79 ^b^	37.60 ± 1.85 ^b^	38.45 ± 2.31 ^b^	38.38 ± 2.32 ^b^
Week 3	24.35 ± 1.03 ^a^	38.95 ± 2.17 ^b^	38.50 ± 1.96 ^b^	39.48 ± 1.08 ^b^	39.10 ± 2.64 ^b^
Week 4	24.28 ± 1.06 ^a^	39.95 ± 1.33 ^b,c^	38.35 ± 2.33 ^b^	41.83 ± 2.17 ^c^	39.58 ± 2.99 ^b^
Feed Intake (g/day)					
Week 1	5.71 ± 0.97 ^a^	11.28 ± 2.18 ^b^	10.08 ± 1.10 ^b^	12.42 ± 1.96 ^b^	10.68 ± 1.48 ^b^
Week 2	6.84 ± 1.65 ^a^	9.00 ± 1.04 ^a,b^	7.82 ± 0.94 ^a,b^	9.67 ± 0.96 ^b^	8.73 ± 2.02 ^a,b^
Week 3	6.89 ± 1.74 ^a^	9.51 ± 0.25 ^b^	7.51 ± 0.58 ^a,b^	9.17 ± 1.05 ^a,b^	8.15 ± 2.60 ^a,b^
Week 4	7.44 ± 0.63 ^a^	9.08 ± 0.56 ^b^	6.82 ± 0.50 ^a,b^	7.80 ± 0.53 ^a,b^	8.20 ± 2.17 ^a,b^
Water Consumption (g/day)					
Week 1	7.05 ± 0.61 ^a^	17.43 ± 10.11 ^b^	16.62 ± 1.80 ^b^	15.07 ± 1.96 ^b^	15.78 ± 1.07 ^b^
Week 2	6.29 ± 0.32 ^a^	17.05 ± 16.36 ^b^	17.25 ± 13.69 ^b^	13.32 ± 0.90 ^b^	16.27 ± 2.06 ^b^
Week 3	6.11 ± 0.14 ^a^	11.85 ± 6.64 ^b^	12.42 ± 5.28 ^b^	12.58 ± 1.33 ^b^	12.03 ± 6.57 ^b^
Week 4	5.74 ± 1.79 ^a^	14.44 ± 10.64 ^b^	14.21 ± 1.14 ^b^	12.10 ± 0.77 ^b^	14.95 ± 10.03 ^b^

Data are expressed as the mean ± SD (*n* = 8 per group). Significant differences are indicated by superscript letters according to ANOVA with post hoc Duncan’s multiple range test. Different superscript letters indicate statistical significance (*p* < 0.05).

**Table 2 metabolites-12-00693-t002:** Plasma lipid profile and hepatic function of *db*/*db* mice supplemented with different doses of SFG for 4 weeks.

			SFG (g/kg)		
	Control	*db*/*db*	1.025	2.05	5.125
TC (mg/dL)	89.3 ± 14.3 ^a^	129.0 ± 25.2 ^b^	108.8 ± 8.8 ^c^	133.0 ± 22.3 ^b,c^	111.4 ± 28.1 ^a,b,c^
TG (mg/dL)	119.1 ± 6.4 ^a^	192.3 ± 21.1 ^c^	158.5 ± 31.0 ^b^	168.8 ± 23.7 ^b^	160.0 ± 21.2 ^b^
AST (U/L)	168.0 ± 59.4 ^a^	300.9 ± 184.1 ^a^	266.9 ± 138.9 ^a^	282.0 ± 218.2 ^a^	202.8 ± 131.7 ^a^
ALT (U/L)	39.1 ± 17.6 ^a^	84.0 ± 27.5 ^b^	93.5 ± 60.8 ^b^	75.5 ± 36.3 ^a,b^	69.9 ± 30.2 ^a,b^

TC, total cholesterol; TG, triacylglycerol; AST, aspartate aminotransferase; ALT, alanine aminotransferase. Data were expressed as the mean ± SD (*n* = 8). Significant differences are indicated by superscript letters according to ANOVA with post hoc Duncan’s multiple range test. Different superscript letters indicate statistical significance (*p* < 0.05).

## Data Availability

The data presented in this study are available from the corresponding author upon reasonable request.

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
