# Peer review of "Improvement of Glycemic Control by a Functional Food Mixture Containing Maltodextrin, White Kidney Bean Extract, Mulberry Leaf Extract, and Niacin-Bound Chromium Complex in Obese Diabetic db/db Mice"

_metabolites, 2022, doi:10.3390/metabo12080693_

Round 1
Reviewer 1 Report
In this manuscript, the authors investigate the potential of Steady-fiber granule in the regulation of blood glucose homeostasis in obese diabetic db/db mice. Steady-fiber granule could be an ideal functional food for the management of type 2 diabetes mellitus in daily life. However, there are many problems with the manuscript. A few major revisions and questions are listed below.
1. Page 3 Line 1-2. The concentration of Steady-fiber granule is unclear.
2. Page 4 Table 1. Is the feed intake and water consumption average per mouse or average per group of mice? It should be explained in more detail in the manuscript.
3. Page 4 Table 1. Why the water consumption of SFG 5.125 group was 162.7 g/day on week 2? Please explain in the manuscript.
4. Page 5 Figure 2B. Please check carefully for significant differences in data between groups. Especially the SFG 2.05 group.
5. Page 7 Table 2. Please unify the abbreviation of aspartate aminotransferase and alanine aminotransferase to avoid the two forms appearing at the same time in one manuscript.
6. The title of Table 2 is “Plasma lipid profile and hepatic function of db/db mice supplemented with different doses of SFG for 30 days”. However, the title of Table 1 is “The body weight, feed intake and water consumption of db/db mice supplemented with different doses of SFG for 4 weeks”. Is 30 day or 4 weeks? Please unified.
Author Response
Reviewer #1
- Page 3 Line 1-2. The concentration of Steady-fiber granule is unclear.
Response: We appreciate the comment by reviewer. SFG was dissolved in distilled water to make 0.5125 g/mL stock solution, which was then diluted to become 0.205 g/mL and 0.1025 g/mL before performing the tests.
- Page 4 Table 1. Is the feed intake and water consumption average per mouse or average per group of mice? It should be explained in more detail in the manuscript.
Response: We appreciate the comment by reviewer. We have revised the description according to the suggestion of reviewer.
Body weight, food intake and water consumption were monitored weekly. Body weight was expressed as average body weight per mouse; food and water intake were calculated as average amount per day per mouse.
- Page 4 Table 1. Why the water consumption of SFG 5.125 group was 162.7 g/day on week 2? Please explain in the manuscript.
Response: We appreciate the comment by reviewer. We apologize for the typo error. It should be 16.27, and we have made the correction.
- Page 5 Figure 2B. Please check carefully for significant differences in data between groups. Especially the SFG 2.05 group.
Response: We appreciate the comment by reviewer. We have rerun the statistical analysis using ANOVA Tukey post hoc test, and made the correction. We have corrected this figure (Figure 2B in original manuscript / Figure 3B in this revised manuscript) according to the suggestion of reviewer.
- Page 7 Table 2. Please unify the abbreviation of aspartate aminotransferase and alanine aminotransferase to avoid the two forms appearing at the same time in one manuscript.
Response: We appreciate the comment by reviewer. We have revised this issue in the Table 2 according to the suggestion of reviewer.
- The title of Table 2 is “Plasma lipid profile and hepatic function of db/db mice supplemented with different doses of SFG for 30 days”. However, the title of Table 1 is “The body weight, feed intake and water consumption of db/db mice supplemented with different doses of SFG for 4 weeks”. Is 30 day or 4 weeks? Please unified.
Response: We appreciate the comment by reviewer. We have revised this issue in the Table 2 according to the suggestion of reviewer.
Reviewer 2 Report
Peer Review on “Improvement of glycemic control by a functional food mixture containing maltodextrin, white kidney bean extract, mulberry leaf extract and niacin-bound chromium complex in obese diabetic db/db mice”.
Manuscript ID: metabolites-1821058
This study on the health benefits of supplementation by steady-fiber granule (SFG) in diabetic mice has some interesting results and is generally well-designed and well-described, showing that glycemic control and insulin sensitivity can be enhanced with this functional food. Further investigations would be worthwhile.
Some comments.
The doses of 1.025, 2.05 and 5.125 g/kg body weight of SFG administered for all experiments except for the lipid and hepatic functions showed that the SFG at 2.05 g/kg BW outperformed the others in all parameters. The reason for this unexpected but very definite and consistent result is not examined. Do you have an explanation? Please address this in your text.
SFG is composed of ingredients which altogether add up to 93.22% of the whole. What is the remaining 6.78% composed of?
Line 1/22. “…at all three doses…” not “…at all three does…”.
Table 1. In the SFG 5.125 column, water consumption, week 2, you say 162.7 g/day.
Line 9/4. The sentence starting with “Given that…” goes nowhere. Delete the words “…and therefore…”.
Author Response
Reviewer #2
- The doses of 1.025, 2.05 and 5.125 g/kg body weight of SFG administered for all experiments except for the lipid and hepatic functions showed that the SFG at 2.05 g/kg BW outperformed the others in all parameters. The reason for this unexpected but very definite and consistent result is not examined. Do you have an explanation? Please address this in your text.
Response: We appreciate the comment by reviewer. The corresponding paragraph has been added ahead of / within the study limitation according to the suggestion of reviewer.
Collectively, multiple mechanisms were proposed to be associated with the bene-fits of the four bioactive ingredients of SFG in regulation of diabetes and obesity. It is of note that the effects of SFG on hyperglycemia and hypertriglyceridemia are not dose-dependent. SFG at 2.05 g/kg BW had favorable effects on most parameters we tested, whereas SFG at 5.125 g/kg BW just ameliorated fasting glucose levels, fructosamine levels and TG contents, whose effects were not better than that of SFG 2.05 group. The benefits of SFG 2.05 group on improving insulin sensitivity in db/db mice were not observed in SFG 5.125 group; further studies are needed to figure out the phenomenon. However, in our experience, few functional foods work in a dose-dependent manner. Willis et. al. also reported that increasing consumption of mixed viscous fiber in hu-man subjects did not influence gut hormone levels, ghrelin and GLP-1, in a dose-dependent manner [53]. They interpreted that different type of fibers may have different impacts on health.
The study has potential limitations. In order to assure accurate and precise dosing, SFG was given to the mice by oral gavage, whose disadvantages include the possibilities of inadvertent tracheal administration and esophageal discomfort. Local inflammation and disturbed absorption may be induced by repeated dosing and then may affect our results. Besides, we only evaluated the effect of SFG for 4 weeks, which might be not long enough to observe the weight-loss effect. Moreover, if we conducted glucose uptake experiments in liver, muscle or adipose tissue or checked β-cell function, it would be better describing the insulin sensitizing effect of SFG and might offer some information to explain the different effects on glycemic control between SFG 2.05 and SFG 5.125 group.
- SFG is composed of ingredients which altogether add up to 93.22% of the whole. What is the remaining 6.78% composed of?
Response: We appreciate the comment by reviewer. The remaining 6.78% is inert excipients to add flavors to the product. We have revised the description in the Methods section for this issue according to the suggestion of reviewer.
- Line 1/22. “…at all three doses…” not “…at all three does…”.
Response: We appreciate the comment by reviewer. We have corrected this typing error according to the suggestion of reviewer.
- Table 1. In the SFG 5.125 column, water consumption, week 2, you say 162.7 g/day.
Response: We appreciate the comment by reviewer. We apologize for the typo error. It should be 16.27 g/day.
- Line 9/4. The sentence starting with “Given that…” goes nowhere. Delete the words “…and therefore…”.
Response: We appreciate the comment by reviewer. We have revised this sentence according to the suggestion of reviewer.
Reviewer 3 Report
The study focused on the effectiveness of supplementation of the diet with functional food containing maltodextrin, white kidney bean extract, mulberry leaf extract and niacin-bound chromium complex on several parameters related to glycemic and lipemic control in animal model. The article covers very important aspects related to the use of functional food in the management of impaired glucose metabolism and diabetes. The paper is original, however, there are several concerns that limit the validity of this work. Please address the following issues:
– The aim of the study provided in lines 19-20 is: to evaluate the potential of SFG in the regulation of blood glucose homeostasis under the obese diabetic condition. However, there is no information about the purpose of the research in the introduction section. Please clearly formulate the purpose of the research so that it corresponds to the title of the article.
– The authors do not provide detailed information about the calculation of the doses of the supplement. It is worth explaining how the doses of the supplement were calculated and whether they were converted to a dose appropriate for humans. The information given in lines 15-71: The doses of SFG were chosen on the basis of our previous studies, and showed no signs of any toxicity tested, is insufficient.
– A flowchart of the study would make it easier to interpret the results. Please include the study flowchart in the methodology.
– Some parts presented in the study results are more the methodology of the study (e.g., lines 2-7 on page 4, lines 5-7 on page 6). I suggest moving these fragments of the text to the research methodology section.
– In the discussion section, the authors do not explain comprehensively the study results, instead, they describe the impact of individual SFG components obtained in studies by other authors. Please interpret the results of your own research in relation to the findings from the literature.
– Study limitations are missing. Please clarify what the study limitations are.
– Finally, the conclusions drawn by the authors go beyond the conducted research (lines 27-29) Taken together, SFG improved glycemic control and insulin sensitivity in db/db mice and could be an ideal functional food for the management of type 2 diabetes mellitus in daily life.
In conclusion, the article requires major revisions concerning interpretation of the results and language quality before being published.
Author Response
Reviewer #3
- The aim of the study provided in lines 19-20 is: to evaluate the potential of SFG in the regulation of blood glucose homeostasis under the obese diabetic condition. However, there is no information about the purpose of the research in the introduction section. Please clearly formulate the purpose of the research so that it corresponds to the title of the article.
Response: We appreciate the comment by reviewer. We have revised the descriptions for this issue according to the suggestion of reviewer.
T2DM is characterized by hyperglycemia resulting from insulin deficiency caused by pancreatic β-cell dysfunction and insulin resistance in insulin-sensitive tissues such as liver, muscle and adipose tissue [6]. While stimulated by elevated blood glucose levels following a meal, insulin secreted by β-cells will work to maintain normal blood glucose mainly by promoting the uptake of glucose in skeletal muscle and adipose tis-sue, as well as by suppressing glucose efflux from liver and lipolysis from adipose. Interaction among all of those organs and tissues involved ensures glucose homeostasis [26]. Impairments in insulin secretion and insulin sensitivity may result in impaired glucose tolerance, leading to T2DM [5]. Thus, improvement of insulin secretion or sensitivity may therefore improve glucose homeostasis, which is beneficial in development and management of T2DM. SFG has wide spectrum of safety manifested in our previous study [27] and its major ingredients, as mentioned above, are all known to have beneficial effects on obesity or diabetes, we then hypothesized that a dietary in-tervention of the functional food SFG would improve the glycemic control and blood lipids under obese diabetic conditions. Accordingly, db/db obese diabetic mice at age of 8 weeks were orally administered with SFG for 4 weeks to observe whether SFG had beneficial effects on obesity and diabetes by attenuating overweight, hyperglycemia, hyperinsulinemia and dyslipidemia in the obese diabetic model.
- The authors do not provide detailed information about the calculation of the doses of the supplement. It is worth explaining how the doses of the supplement were calculated and whether they were converted to a dose appropriate for humans. The information given in lines 15-71: The doses of SFG were chosen on the basis of our previous studies, and showed no signs of any toxicity tested, is insufficient.
Response: We appreciate the comment by reviewer. We have revised the descriptions for this issue according to the suggestion of reviewer.
The doses of SFG were chosen on the basis of our previous studies, and showed no signs of any toxicity tested [27]. In brief, the recommended dosage of SFG for human is 1 serving (5 g) to 2 servings per day. Based on the body surface area, converting human dose to equivalent mouse dose is multiplying the human dose by 12.3 [30]. Therefore, the doses for mice were set at 1.025 g/kg as 1-fold dose, 2.05 g/kg as 2-fold dose (high recommended dose), and 5.125 g/kg as 5-fold dose.
- A flowchart of the study would make it easier to interpret the results. Please include the study flowchart in the methodology.
Response: We appreciate the comment by reviewer. We have added a flowchart of the study in Figure 1 of this revised manuscript according to the suggestion of reviewer.
- Some parts presented in the study results are more the methodology of the study (e.g., lines 2-7 on page 4, lines 5-7 on page 6). I suggest moving these fragments of the text to the research methodology section.
Response: We appreciate the comment by reviewer. We have revised the descriptions for this issue in this revised manuscript according to the suggestion of reviewer.
- In the discussion section, the authors do not explain comprehensively the study results, instead, they describe the impact of individual SFG components obtained in studies by other authors. Please interpret the results of your own research in relation to the findings from the literature.
Response: We appreciate the comment by reviewer. We have added the descriptions in the Discussion for this issue in this revised manuscript according to the suggestion of reviewer.
Collectively, multiple mechanisms were proposed to be associated with the bene-fits of the four bioactive ingredients of SFG in regulation of diabetes and obesity. It is of note that the effects of SFG on hyperglycemia and hypertriglyceridemia are not dose-dependent. SFG at 2.05 g/kg BW had favorable effects on most parameters we tested, whereas SFG at 5.125 g/kg BW just ameliorated fasting glucose levels, fructosamine levels and TG contents, whose effects were not better than that of SFG 2.05 group. The benefits of SFG 2.05 group on improving insulin sensitivity in db/db mice were not observed in SFG 5.125 group; further studies are needed to figure out the phenomenon. However, in our experience, few functional foods work in a dose-dependent manner. Willis et. al. also reported that increasing consumption of mixed viscous fiber in human subjects did not influence gut hormone levels, ghrelin and GLP-1, in a dose-dependent manner [53]. They interpreted that different type of fibers may have different health impacts.
- Study limitations are missing. Please clarify what the study limitations are.
Response: We appreciate the comment by reviewer. We have added the study limitation in the Discussion of this revised manuscript according to the suggestion of reviewer.
The study has potential limitations. In order to assure accurate and precise dosing, SFG was given to the mice by oral gavage, whose disadvantages include the possibilities of inadvertent tracheal administration and esophageal discomfort. Local inflammation and disturbed absorption may be induced by repeated dosing and then may affect our results. Besides, we only evaluated the effect of SFG for 4 weeks, which might be not long enough to observe the weight-loss effect. Moreover, if we conducted glucose uptake experiments in liver, muscle or adipose tissue or checked β-cell function, it would be better describing the insulin sensitizing effect of SFG and might offer some information to explain the different effects on glycemic control between SFG 2.05 and SFG 5.125 group.
- Finally, the conclusions drawn by the authors go beyond the conducted research (lines 27-29) Taken together, SFG improved glycemic control and insulin sensitivity in db/db mice and could be an ideal functional food for the management of type 2 diabetes mellitus in daily life.
Response: We appreciate the comment by reviewer. We have revised the descriptions for this issue in the Abstract section of this revised manuscript according to the suggestion of reviewer.
Round 2
Reviewer 1 Report
The authors have addressed all my comments in this revision.
Reviewer 3 Report
I accept the revisions provided by the authors.
I have a comment regarding the conclusions:
The conclusions provided in lines 361-363 need to be corrected, as they go beyond the received results “supplementation of functional food SFG, to meet the recommended daily intake of RS, is therefore an ideal option for T2DM patients to manage the progression of the disease in their daily life…”
Please submit the same conclusions as in the abstract: “These findings suggest that SFG improves glycemic control and insulin sensitivity in db/db mice and can be available as an option for functional foods to aid in management of type 2 diabetes mellitus in daily life.”